# A Novel and Green Method for Preparing Highly Conductive PEDOT:PSS Films for Thermoelectric Energy Harvesting

**DOI:** 10.3390/polym16020266

**Published:** 2024-01-18

**Authors:** Fuwei Liu, Luyao Gao, Jiajia Duan, Fuqun Li, Jingxian Li, Hongbing Ge, Zhiwei Cai, Huiying Li, Mengke Wang, Ruotong Lv, Minrui Li

**Affiliations:** 1College of Physics and Electronic Engineering, Xinyang Normal University, Xinyang 464000, China; 2Key Laboratory of Advanced Micro/Nano Functional Materials of Henan Province, Xinyang Normal University, Xinyang 464000, China; 3Energy-Saving Building Materials Innovative Collaboration Center of Henan Province, Xinyang Normal University, Xinyang 464000, China

**Keywords:** PEDOT:PSS, water treatment, thermoelectrics

## Abstract

As a π-conjugated conductive polymer, poly(3,4-ethylenedioxythiophene):poly(styrene sulfonate) (PEDOT:PSS) is recognized as a promising environmentally friendly thermoelectric material. However, its low conductivity has limited applications in the thermoelectric field. Although thermoelectric efficiency can be significantly enhanced through post-treatment doping, these processes often involve environmentally harmful organic solvents or reagents. In this study, a novel and environmentally benign method using purified water (including room temperature water and subsequent warm water) to treat PEDOT:PSS film has been developed, resulting in improved thermoelectric performance. The morphology data, chemical composition, molecular structure, and thermoelectric performance of the films before and after treatment were characterized and analyzed using a scanning electron microscope (SEM), Raman spectrum, XRD pattern, X-ray photoelectron spectroscopy (XPS), and a thin film thermoelectric measurement system. The results demonstrate that the water treatment effectively removes nonconductive PSS from PEDOT:PSS composites, significantly enhancing their conductivity. Treated films exhibit improved thermoelectric properties, particularly those treated only 15 times with room temperature water, achieving a high electrical conductivity of 62.91 S/cm, a Seebeck coefficient of 14.53 μV K^−1^, and an optimal power factor of 1.3282 µW·m^–1^·K^–2^. In addition, the subsequent warm water treatment can further enhance the thermoelectric properties of the film sample. The underlying mechanism of these improvements is also discussed.

## 1. Introduction

With growing concerns about the environment and energy, there is increasing demand for renewable material-based energy technologies [1,2,3]. In particular, energy conversion materials that harness sustainable energy from the surrounding environment are garnering significant interest [4]. Thermoelectric materials, which facilitate the direct conversion of heat energy to electric power through the movement of charge carriers in solids, are seen as having wide-ranging applications in sustainable and clean energy [5,6]. The thermoelectric conversion efficiency of a material is primarily associated with its Seebeck coefficient (S), electrical conductivity (σ), thermal conductivity (κ), and absolute temperature (T), evaluated by the dimensionless thermoelectric figure of merit, ZT = S^2^σT/κ. Excluding thermal conductivity, a parameter named power factor (represented as PF = S^2^σ) is another metric for assessing thermoelectric performance.

Inorganic thermoelectric materials, known for their high output power density, have been extensively studied. However, their high cost and fragility limit their application in flexible, portable, and wearable devices. Conversely, organic thermoelectric materials, marked by their low density, excellent flexibility, abundant resources, and inherent low thermal conductivity, have recently been the focus of numerous studies. Among various conducting polymers, PEDOT:PSS stands out for its high energy conversion efficiency, making it an ideal organic TE material. For instance, Xu et al. prepared a doped PEDOT:PSS aqueous solution using acids and alkalis, investigating the effect of pH variation on TE performance. Their results indicated that decreasing the pH value enhances TE properties, achieving an optimal power factor of 1.35 μW m^−1^ K^−2^ [7]. Kumar et al. enhanced the wettability of PEDOT:PSS on a glass substrate using methanol, followed by treatment with 1 M sulfuric acid, simultaneously increasing its electrical conductivity and Seebeck coefficient [8]. Similarly, sulfuric acid vapor was also employed to treat PEDOT:PSS thin films, and the obtained samples presented significantly enhanced electrical conductivity with an improved Seebeck coefficient, yielding a maximum power factor of 17.0 µW·m^–1^·K^–2^ [9]. Moreover, the TE properties of PEDOT:PSS films can be significantly enhanced by adding, during and/or post treatment, organic acids [10,11], organic solvents [12,13,14,15], and other chemical reagents (such as surfactants [16,17], salts solutions [18], and ionic liquids [19,20]). Additionally, incorporating inorganic fillers with a high conductivity or Seebeck coefficient has proven effective in enhancing TE properties. For example, Du et al. fabricated highly conductive CNTs/PEDOT:PSS composite films through a DMSO treatment, achieving a power factor of 108.7 µW·m^−1^·K^−2^ [21]. Chen et al. developed layer-like PEDOT:PSS/SWCNT composite films with honeycomb-like structures, which achieved a high power factor value of around 45.72 µW·m^–1^·K^–2^ after H_2_SO_4_ treatment [22]. See et al. created a novel composite film composed of Te nanorods and PEDOT:PSS, demonstrating an exceptional power factor of approximately 70 µW·m^–1^·K^–2^ [23]. These findings are heartening and suggest further research is warranted. 

Despite these advancements, the use of toxic solvents and fillers, along with complex preparation processes, poses environmental concerns. Therefore, simple, green approaches are urgently needed. This study employs purified water to treat PEDOT:PSS films through a straightforward washing process. As is well known, water is rich in resources, and the preparation of DI water is relatively simple, convenient, and low-cost. Notably, this treatment can also selectively remove nonconductive PSS and effectively modify the crystallinity of PEDOT molecules. The electrical conductivity of the samples significantly increases with the water washing cycle number, reaching 76.35 S cm^–1^ after 20 cycles. Simultaneously, the Seebeck coefficient decreases marginally. Continuous optimization with warm water yields a high room temperature power factor of approximately 4.4344 µW·m^–1^·K^–2^, which is 672 times larger than that of the pristine sample. The mechanism of this significant enhancement of TE properties due to the water washing process is also examined.

## 2. Materials and Methods

### 2.1. Materials

A commercial PEDOT:PSS aqueous solution (PH1000, Mw = 326.388) was procured from Heraeus Company (Hanau, Germany). Deionized (DI) water with a relatively high resistance of approximately 18.2 MΩ cm was used to dilute the PEDOT:PSS solution during film preparation and to treat film samples during the water washing process. 

### 2.2. Pre-Treatment and Film Preparation

Initially, the acquired PH1000 was diluted with DI water to a concentration of 6 mg/mL PEDOT:PSS solution. Subsequently, 150 µL of this solution was drop-coated onto a pre-cleaned glass substrate (1.2 × 1.2 cm^2^) and allowed to dry under ambient conditions. The as-prepared PEDOT:PSS films were then dipped into DI water (around 17 °C) for 10 s, removed, and dried on a heating platform at 60 °C. This immersion and drying procedure was repeated 20 times. PEDOT:PSS film samples treated 5, 10, 15, and 20 times were selected for further characterizations. These as-prepared films were labelled as PH-5, PH-10, PH-15, and PH-20, respectively. To further improve its performance, warm water (30 °C) was used in the subsequent water washing process. The immersion and drying procedure was repeated until the electrical conductivity reached a relatively stable value (around 60 times).

### 2.3. Characterizations and Measurements

Film morphologies were obtained by employing a field emission scanning electron microscope (FE-SEM, S-4800, Hitachi Limited, Tokyo, Japan). Film thickness was determined using a step profiler (ET-4000M, Kosaka Laboratory Ltd., Tokyo, Japan). Molecular structure changes in the films were analyzed using a Nicolelis5 Fourier transform infrared spectroscopy (FTIR) instrument (ThermoFisher, Waltham, MA, USA) scanning in the range of 400~4000 cm^–1^. Raman spectroscopy was conducted with a LabRAM HR Evolution instrument using an 514 nm excitation laser (HORIBA, Kyoto, Japan). XRD spectra were obtained with a Smartlab 9 (Rigaku Corporation, Tokyo, Japan) X-ray Diffractometer, using a scanning range 2*θ* = 5°–60°. Electron-binding energies of the films were measured using a Thermo KAlpha X-ray Photoelectron Spectroscope (XPS, Thermo Fisher Scientific). The UV-Vis absorption spectra were collected using a LAMBDA950 UV/Vis/NIR Spectrophotometer (PerkinElmer, Waltham, MA, USA). The conductivity and Seebeck coefficient of PEDOT:PSS films were tested using a MRS-3 thin-film thermoelectric tested system (Wuhan Joule Yacht Science & Technology Co., Ltd., Wuhan, China). 

## 3. Results and Discussion

### 3.1. Fabrication of PEDOT:PSS Film and the Water Treatment

Figure 1 displays the schematic illustration of the water treatment process for the PEDOT:PSS film. A typical drop coating method was used to prepare the pristine PEDOT:PSS film. After drying at room temperature, the prepared films were soaked in DI water to selectively remove PSS and improve the molecular arrangement of PEDOT. Due to the solubilizing effect of the water on the film, the soaking duration was limited to 10 s, followed by a drying process. To thoroughly understand the impact of the water on the film, the immersion and drying steps were repeated multiple times. Notably, the water treatment led to the formation of an interconnected porous network. The effective removal of PSS suggests that the resultant films would exhibit enhanced electrical conductivities and thermoelectric properties, which are discussed subsequently.

### 3.2. Structure Characterization and Analysis

The films were examined using SEM, with the results presented in Figure 2. The pristine film displayed a relatively compact structure. In contrast, water treatment significantly altered the film’s cross-sectional morphology, revealing sheet-like structures with microporous morphology. Notably, the film thickness increased following the water treatment, aligning with the measurements obtained via the step profiler. The emergence of porous structures and increased film thickness can be attributed to DI water penetration and the resultant microstructural changes. The film surfaces became non-uniform due to PEDOT:PSS dissolution in water (Figure 2f). Despite these changes, all water-treated films exhibited highly interconnected microporous structures within the PEDOT:PSS, offering efficient channels for carrier transportation. 

Appendix A presents the FTIR spectra of the samples before and after different cycles of the water washing treatment. In the original PEDOT:PSS film, the transmission peaks centered at 1161 cm^–1^ and 1130 cm^–1^ are assigned to the S–O of PSS; the peak observed at 1011 cm^–1^ is related to the S-phenyl stretching of PSS. Peaks centered at 856 cm^–1^ and 943 cm^–1^ correspond to the C–S stretching of PEDOT, while peaks located at 1261 cm^–1^, 1371 cm^–1^, and 1060 cm^–1^ are ascribed to C–O–C, C–C, and EDOT ring structures, respectively [24]. It can be seen that the peaks associated with PSS at 1161 cm^–1^ and 1130 cm^–1^ decreased slowly in intensity with water treatments. To quantify this variation, a ratio of *A*_1130_/*A*_1060_ was defined as the ratio of the area of the PSS peak centered at 1130 cm^–1^ to the area of the PEDOT peak centered at 1060 cm^–1^. The calculated values are presented in Appendix A and a declining trend was observed, indicating the removal of PSS from PEDOT:PSS. It is also noteworthy that a characteristic peak of the C=C quinoidal structure in PEDOT appeared at 1450 cm^–1^ after 15 cycles of water washing, and became more apparent after 20 cycles of the water treatment. The observed changes in PEDOT:PSS contribute to the improvement of the crystallinity of the polymer, thus increasing its electrical conductivity.

As a supplement to the infrared spectra, Raman spectroscopy, a nondestructive testing method, was continuously employed to identify structural variations in the organic polymers and carbon-based materials [25,26,27]. Figure 3a displays the Raman spectra of the PEDOT:PSS films including the pristine as-prepared one and those treated with different cycles of the water washing treatment. Characteristic peaks associated with PEDOT:PSS structures were observed, with the peak attributions summarized in Table 1 [28,29]: peaks centered at 1566 cm^–1^ and 1493 cm^–1^, corresponding to asymmetric C_α_=C_β_ stretching, became increasingly prominent after the water treatment; the peak at 1428 cm^–1^, attributed to symmetric C_α_=C_β_(–O) stretching vibrations of the five-membered thiophene ring, became sharper and more intense. Similar trends were observed at 1367 cm^–1^ (C_β_–C_β_ stretching), 1254 cm^–1^ (inter-ring C_α_–C_α_ stretching), 1117 cm^–1^ (C–C ring bending vibration), 1094 cm^–1^ (C–O–C deformation), 701 cm^–1^ (symmetric C–S–C deformation), 437 cm^–1^ (SO_2_ bending), 988 cm^–1^, and 576 cm^–1^ (deformation of oxyethylene ring). The peak centered at 522 cm^–1^, indicating the presence of the PSS component, exhibited a significant reduction in intensity after the water treatment, suggesting the effective removal of the PSS component. To quantify these changes, a ratio of *I*_522_/*I*_988_ was defined as the ratio of the intensity of the PSS peak centered at 522 cm^–1^ to the intensity of the PEDOT peak located at 988 cm^–1^ [27]. The calculated values are displayed in Appendix A and a declining trend was observed, further confirming the above assertion. The peak related to the stretching vibration of symmetric C_α_=C_β_(–O) at 1425 cm^–1^ exhibited a red shift, similar to treatments using dielectric solvents. Generally, shifts in symmetric C_α_=C_β_(–O) stretching are influenced by the relative ratio of benzene and quinoid rings, correlating with the oxidation degree of the molecular structures and/or the configuration of the PEDOT molecule. In a quinoid-dominated structure, conjugated π-electrons are more easily delocalized across the entire PEDOT chain, with adjacent thiophene rings nearly parallel to one another, thereby enhancing charge carrier mobility compared to a coiled benzoid-dominated structure. Additionally, after the water treatment a significant increase in corresponding band intensities was observed, possibly due to PSS removal and/or the reorienting effect of water on PEDOT chains. Further details regarding the variation of PEDOT:PSS films during treatment are discussed in subsequent sections.

X-ray diffraction (XRD) patterns were analyzed to further comprehend the impact of the water treatment on PEDOT:PSS structures, with results displayed in Figure 3b. The PEDOT:PSS film without water washing revealed two distinct diffraction peaks at 2*θ* = 26.4° and 18.5°, corresponding to the interchain planar π-π stacking distance (d_(010)_) of aromatic rings in PEDOT and the amorphous halo of PSS, respectively [30,31]. Post water treatment, the intensity of the PSS peaks noticeably decreased, indicating either the effective removal of PSS or increased conformational disorder. Conversely, the peak related to the planar ring-stacking of PEDOT showed enhanced intensity, signifying increased lamellar stacking and improved crystallinity. This ordered and oriented pattern with relatively high reflective intensities should facilitate carrier transport in the continuous porous framework, consequently boosting electrical conductivity. These alterations are attributed to the selective removal of nonconductive PSS and subsequent molecular rearrangement in PEDOT. It is worth mentioning that these structural changes are analogous to the results reported in those studies [13]. 

UV-Vis-NIR spectra were also applied to investigate the compositional and conformational variation in PEDOT:PSS structures. No noticeable changes can be observed during the water washing process, until the cycle of treatments reached 20. For analytic simplicity, the data from the pristine PEDOT:PSS were compared with those from the film treated 20 times, and the results are presented in Appendix A. As depicted, the absorption peak at 256 nm can be assigned to the aromatic ring of PSS [24,29]. After a water treatment for 20 cycles, this peak decreased in density, clear evidence of PSS removal. However, the two curves almost coincide in the visible and NIR regions, demonstrating no obvious variation of the oxidation degree of the molecular structures [32]. 

To confirm the compositional changes in the treated films further, an X-ray electron spectroscopy (XPS) analysis was also performed, and its results are presented in Figure 4. The elements C, O, and S were identified in the full-range XPS spectra. The reduction in the insulating PSS content from PEDOT:PSS films during the H_2_O treatment was confirmed by analyzing the S2p spectrum in the XPS. As illustrated in Figure 4b–f, the binding energies between 166 and 172 eV corresponded to sulfur atoms S2p bands in PSS, while the ones between 162 and 166 eV were related to the S2p bands of sulfur in PEDOT [33]. Moreover, PEDOT bands featured two peaks at approximately 164 eV and 165 eV for S2p_3/2_ and S2p_1/2_ thiophene sulfur, while two peaks at around 168 eV and 169 eV corresponded to the S2p_3/2_ and S2p_1/2_ sulfur of PSS. Based on the changes in peak areas corresponding to PSS and PEDOT, the structural modifications of the films were inferred. Calculations revealed that the PSS/PEDOT ratio decreased from 3.02 to 2.84, 2.42, 2.13, and 1.55 after 5, 10, 15, and 20 water washing cycles, respectively. This reduction can be ascribed to the conformational changes resulting from the removal of PSS during the H_2_O washing process, indicating that PSS was progressively eliminated with an increasing number of H_2_O treatments. The selective removal of the nonconductive-phase PSS effectively enhanced the conductivity of the PEDOT:PSS films. Additionally, the XPS results were consistent with the observed increase in electrical conductivity.

### 3.3. Thermoelectric Performance of PEDOT:PSS Film Prepared via Water Treatment

In the experiments, the pristine PEDOT:PSS films were prepared using a drop coating method, and the obtained samples exhibited a relatively low conductivity of 0.25 S cm^–1^ (Figure 5a), aligning with other research reports and attributed to the high content of PSS in the original films. After water washing, film conductivity increased substantially. For instance, conductivity rose to 26.88 S cm^–1^ after 5 wash cycles, and to 76.35 S cm^–1^ after 20 cycles, representing a nearly threefold increase. The ratio of PSS to PEDOT decreased from 3.02 to 1.55 over 20 wash cycles, indicating the effective removal of PSS and the resultant increase in carrier concentration in the films. Enhanced peak intensity in the XRD patterns of PEDOT’s inter-chain planar ring stacking suggests improved crystallization, which is generally known to enhance the charge transfer within and between chains, thereby improving polymer conductivity. As shown in Figure 2, a porous structure was generated during the water washing process, which may partly reduce the number of the conductive pathways. Notably, as discussed above, networks for charge transport have become more efficient following the water washing treatment. As a result, the obtained films showed much higher electric conductivities when compared to untreated ones.

Figure 5b illustrates the Seebeck coefficient variation in PEDOT:PSS films following different water treatment cycles. The untreated film maintained a Seebeck coefficient of ~16.34 µV K^–1^, which gradually decreased to 16.10, 15.24, 14.53, and 11.73 µV K^–1^ after 5, 10, 15, and 20 wash cycles, respectively. This decrease likely correlates with the marked increase in carrier concentration due to PSS reduction. Figure 5c displays the power factor variation, showing that the pristine PEDOT:PSS film had a power factor of 0.0066 µW·m^–1^·K^–2^ due to low conductivity. Post water treatment for 5, 10, 15, and 20 cycles, the power factors reached 0.6971, 1.3473, 1.3282, and 1.0514 µW·m^–1^·K^–2^, respectively. During heating, the power factor of the thin film exhibited an upward trend, reaching 1.7986 µW·m^–1^·K^–2^ at 360 K after 20 wash cycles, which is at least 200 times higher than that of the pristine film. This upward trend of the power factor with temperature may be associated with the changes in the PEDOT:PSS structure, such as a transformation from a benzenoid type to a quinoidal one and the subsequent increase in the crystallinity [26,34]. 

Certainly, this method still has the potential for further developments. We also explored the relevant strategies in the experimental part, including repeating the water treatment at room temperature (water temperature remains at around 17 °C) for the first 20 times, with a subsequent warm water (30 °C) washing process for further modification. The results are shown in Appendix A. During the warm water washing process, the conductivities of the samples continued their previous incremental growth state until they reached a maximum of 221.52 S cm^−1^ after 60 cycles of further treatment (these conductivity values in Appendix A were also confirmed by the film resistance shown in Appendix A). This conductivity was comparable to or even higher than most films modified using polar solvents, such as EG, DMSO, THF, etc. For example, PEDOT:PSS films doped with THF (25 vol%), DMF (25 vol%), and DMSO (25 vol%) possessed conductivities of ~4 S cm^−1^, 30 S cm^−1^, and 80 S cm^−1^, respectively [12]. Additionally, the electrical conductivity could be raised to 3.5 S cm^−1^ when 20% of the film was EG [35]. This great enhancement in conductivity may be mainly induced by a significant structural change in PEDOT:PSS (e.g., a better crystallinity of PEDOT and/or more ordered molecular arrangement), rather than the removal of a large amount of PSS component (Appendix A). It is also worth noting that the Seebeck coefficient has been kept at around 14 µV K^–1^, which is very similar to that of the sample treated using DMSO (~15 µV K^–1^) [36]. Therefore, an optimal power factor of 4.4344 µW·m^–1^·K^–2^ can be achieved, which is three times higher than that of the sample treated only using room temperature water (a PF value of 1.3282 μW·m^−1^·K^−2^, as reported in this work before warm water washing), 672 times larger than that of the pristine sample (0.0066 μW·m^−1^·K^−2^). More than that, the value of the power factor achieved is comparable to or even higher than those of some composites, such as PEDOT:PSS/Ca_3_Co_4_O_9_ composite films (3.77 µW·m^–1^·K^–2^) [37], some multilayered film systems like PEDOT:PSS/PANI (3.0 µW·m^–1^·K^–2^) [38], and PEDOT:PSS/P3HT (5.79 µW·m^–1^·K^–2^) [39]. Although the power factor is not as good as that of the H_2_SO_4_-treated samples (as reported in our previous work, a high power factor of around 47 µW·m^–1^·K^–2^ can be obtained because of the strong effect of concentrated sulfuric acid on the polymer structure, including the removal of PSS, better crystallinity of the conductive PEDOT:PSS, and variation of the microstructures) [29], the process of water treatment was very environmentally friendly, and no harmful reagent was used. 

Another crucial factor that affects the thermoelectric properties of TE materials is thermal conductivity. In general, most conducting polymers, including PEDOT:PSS, possess inherent low thermal conductivities, which remain in a narrow range of 0.1–0.4 W·m^−1^·K^−1^, substantially lower (around 2–4 orders of magnitude) than those of inorganic TE materials [40]. For example, a DMSO-doped PEDOT:PSS film showed a thermal conductivity of 0.15 W·m^−1^·K^−1^, and the original sample showed a thermal conductivity of 0.17 W·m^−1^·K^−1^ [36,41]. It is worth noting that the introduction of an inorganic material would lead to an increase in thermal conductivity to a certain extent. In our experiment, no inorganic component was incorporated, and during the water washing process only a portion of the PSS component was removed. Considering that the PSS component is essentially a polymer, it may have little effect on the variation of the thermal conductivity. In addition, after multiple cycles of water treating, the PEDOT:PSS films present porous structures (as shown in Figure 2), which can effectively promote the scattering of photons, thus reducing their thermal conductivity without significant influence on their electrical performance. Therefore, it is reasonable to speculate that the thermal conductivities of all samples prepared in this work are kept at a relatively low level.

## 4. Conclusions

In conclusion, a green and innovative method was developed for treating PEDOT:PSS films with water to enhance their thermoelectric (TE) properties. Structural characterization confirmed the selective removal effect of water on PSS, promoting the formation of ordered PEDOT molecular chains and a continuous conductive network, thereby enhancing electrical conductivities. By varying the number of water treatments, PEDOT:PSS films achieved a high electrical conductivity of up to 76.35 S cm^–1^, with only a slight decline in the Seebeck coefficient. With 20 water wash cycles, the film’s power factor reached a high value of 1.0514 µW·m^–1^·K^–2^, escalating to 1.7986 µW·m^–1^·K^–2^ at 360 K, indicating its potential as a TE material. When further treating the film with warm water (60 cycles), a higher conductivity of 221.52 S cm^−1^ and a Seebeck coefficient of 14.15 µV K^–1^ can be achieved. As a result, the power factor reached up to 4.4344 µW·m^–1^·K^–2^ at room temperature. This study presents a straightforward, green, and efficient method for improving the conductivity and TE performance of PEDOT:PSS films, paving the way for expanding their applications in various fields.

## Figures and Tables

**Figure 1 polymers-16-00266-f001:**
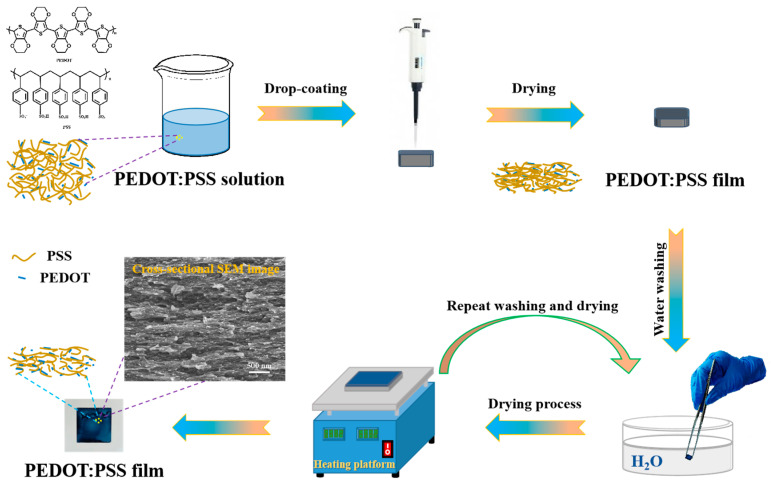
Schematic representation of the formation mechanism of water-treated PEDOT:PSS film.

**Figure 2 polymers-16-00266-f002:**
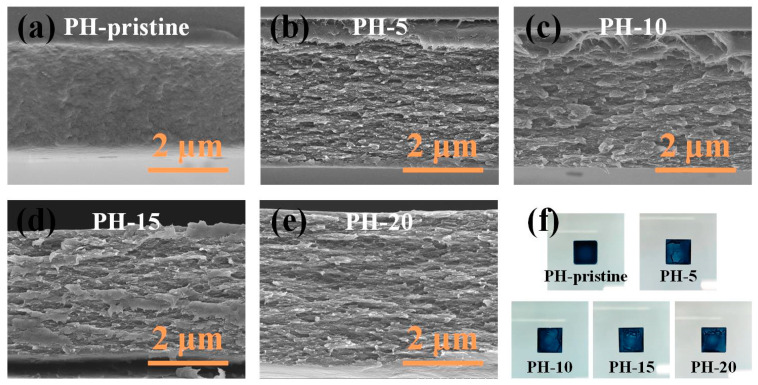
Cross-section SEM images of (**a**) pristine PEDOT:PSS film; (**b**) film water treated 5 times; (**c**) film water treated 10 times; (**d**) film water treated 15 times; (**e**) film water treated 20 times. (**f**) Optical photograph of all samples.

**Figure 3 polymers-16-00266-f003:**
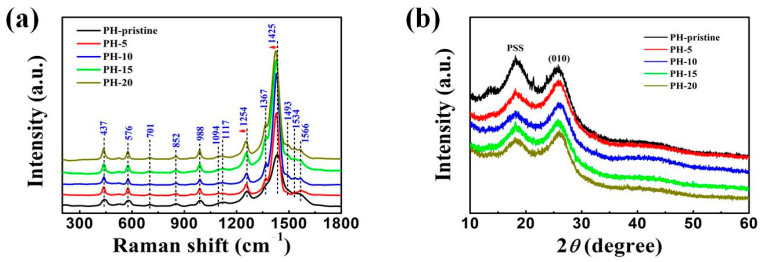
Variations in PEDOT:PSS structures before and after different cycles of water washing treatment: (**a**) Raman spectra; (**b**) XRD spectra.

**Figure 4 polymers-16-00266-f004:**
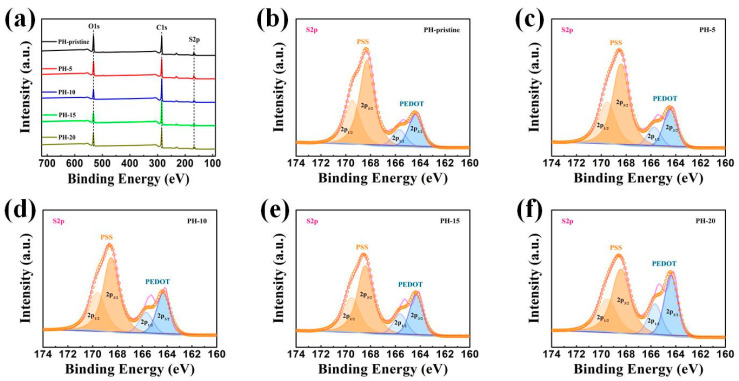
XPS spectra of PEDOT:PSS films with and without water treatment: (**a**) full-scale spectra of PEDOT:PSS; (**b**) S2p spectra of pristine PEDOT:PSS film; (**c**) S2p spectra of PEDOT:PSS film washed 5 times; (**d**) S2p spectra of PEDOT:PSS film washed 10 times; (**e**) S2p spectra of PEDOT:PSS film washed 15 times; (**f**) S2p spectra of PEDOT:PSS film washed 20 times.

**Figure 5 polymers-16-00266-f005:**
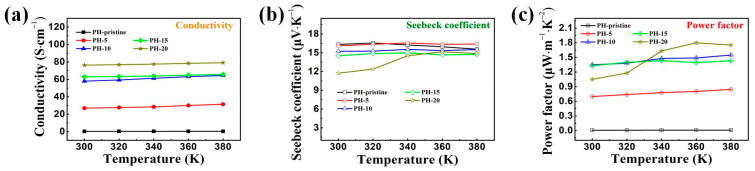
Variation of the TE performance of PEDOT:PSS film before and after treatment using room temperature water: (**a**) conductivity, (**b**) Seebeck coefficients, and (**c**) power factor.

**Table 1 polymers-16-00266-t001:** The characteristic Raman bands and their assignments for pristine PEDOT:PSS film.

Raman Bands (cm^−1^)	Assignments
1566, 1493	asymmetrical Cα=Cβ vibrations
1428	symmetric Cα=Cβ(–O) stretching vibrations of the five-membered thiophene ring
1367	Cβ–Cβ stretching
1254	Cα–Cα inter ring stretching
1117	bending vibration of C–C ring
1094	C–O–C stretching
988	deformation of the oxyethylene ring
701	deformation of the symmetric C–S–C
576	deformation of the oxyethylene ring
437	SO_2_ bending

## Data Availability

The data that support the findings of this study are available from the corresponding author.

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
