# Peer review of "A Novel and Green Method for Preparing Highly Conductive PEDOT:PSS Films for Thermoelectric Energy Harvesting"

_polymers, 2024, doi:10.3390/polym16020266_

Round 1

Reviewer 1 Report

Comments and Suggestions for Authors
  1. (1) The authors have introduced an innovative and environmentally friendly method for crafting thermoelectric energy harvesters. To highlight the superiority of the proposed method, it is recommended to conduct a comparative analysis with conventional thermoelectric energy harvesters.

(2) The paper employs a 'water treatment' approach to eliminate PSS and enhance electrical conductivity. While this method is straightforward, similar techniques have been documented in numerous prior studies. Therefore, it is highly advisable to underscore the novelty of the fabrication methods introduced in this paper.

(3) In Figure 3 (Raman spectra), the relative intensity of the main peaks can provide support for understanding the composition or decomposition of two different materials. A comprehensive discussion in the relevant section, referring to insights from the paper (DOI: 10.1016/j.jmrt.2022.02.134), is strongly encouraged.

(4) The manuscript primarily focuses on observing conductivity, Seebeck coefficient, and power factor. However, these values alone may not suffice to gauge the energy harvesting performance of the fabricated samples. Therefore, it is suggested to incorporate additional experimental results in the revised manuscript.

Comments on the Quality of English Language
  1. (1) The authors have introduced an innovative and environmentally friendly method for crafting thermoelectric energy harvesters. To highlight the superiority of the proposed method, it is recommended to conduct a comparative analysis with conventional thermoelectric energy harvesters.

(2) The paper employs a 'water treatment' approach to eliminate PSS and enhance electrical conductivity. While this method is straightforward, similar techniques have been documented in numerous prior studies. Therefore, it is highly advisable to underscore the novelty of the fabrication methods introduced in this paper.

(3) In Figure 3 (Raman spectra), the relative intensity of the main peaks can provide support for understanding the composition or decomposition of two different materials. A comprehensive discussion in the relevant section, referring to insights from the paper (DOI: 10.1016/j.jmrt.2022.02.134), is strongly encouraged.

(4) The manuscript primarily focuses on observing conductivity, Seebeck coefficient, and power factor. However, these values alone may not suffice to gauge the energy harvesting performance of the fabricated samples. Therefore, it is suggested to incorporate additional experimental results in the revised manuscript.

Reviewer 2 Report

Comments and Suggestions for Authors

The manuscript by Liu et al. describes a novel method to achieve high conductivity PEDOT:PSS film using water treatment. The manuscript in its current version can not be accepted for publication. There are some fundamental flaws in designing the experiments and several results are not properly discussed. Some of my concerns are listed below:

(i) Why did the authors stop after 20 times water washing for PEDOT: PSS film formation? It seems the conductivity was increasing continuously with water treatment. What will happen if the washing cycle is more than 20 times? The authors should have investigated an optimized washing methodology to achieve the best performance.

(ii) It is not well understood why the pores are helping in improved conductivity. With higher water treatment, PSS are cleaned as a result the pores are being formed. The pores should disturb the continuous charge transport. The authors should have discussed in detail the exact conductivity mechanism in charge transport.

(iii) In Figure 5c, the power factor of 20 times washed showed a completely different temperature-dependent trend compared to all other samples. Why there was a sudden jump around 330 degrees Celsius? The authors should have discussed this elaborately.

(iv) The thermoelectric power reported with this method is much smaller compared to some of the reported methods of using different solvents like DMSO. Can the author enlighten the reason behind such a lesser performance with water treatment compared to other solvent treatments?   

Round 2

Reviewer 1 Report

Comments and Suggestions for Authors

The authors revised the manuscript based on the reviewer's comments. Thus, the reviewer thought that this manuscript can be published in this journal.

Comments on the Quality of English Language

The authors revised the manuscript based on the reviewer's comments. Thus, the reviewer thought that this manuscript can be published in this journal.

Author Response

Thanks a lot for your constructive comments and suggestions,which are all valuable and very helpful for improving the quality of our paper. 

Reviewer 2 Report

Comments and Suggestions for Authors

The authors' response to my first comment is still not satisfactory. The authors should have performed process optimization for the washing. The authors should have tried how the thermoelectric power varies with the washing cycle more than 20 times. The reported thermo-electric efficiency is also not so impressive. I do not recommend the publication of the article.

Round 3

Reviewer 2 Report

Comments and Suggestions for Authors

I have seen the authors have significantly improved the manuscript with systematic study and discussions backed by experimental results. I recommend the publication of the article.